# A Comprehensive Study of CsSnI_3_-Based Perovskite Solar Cells with Different Hole Transporting Layers and Back Contacts

**DOI:** 10.3390/mi14081562

**Published:** 2023-08-06

**Authors:** Seyedeh Mozhgan Seyed-Talebi, Mehrnaz Mahmoudi, Chih-Hao Lee

**Affiliations:** 1Department of Engineering and System Science, National Tsing Hua University, Hsinchu 300044, Taiwan; 2Department of Chemistry, Faculty of Science, Shahid Rajaee Teacher Training University, Tehran 1678815811, Iran; mahmudi.mehr1981@gmail.com

**Keywords:** CsSnI_3_, hole transporting layer (HTL), perovskite solar cell (PSC), SCAPS-1D software

## Abstract

By an abrupt rise in the power conservation efficiency (PCE) of perovskite solar cells (PSCs) within a short span of time, the instability and toxicity of lead were raised as major hurdles in the path toward their commercialization. The usage of an inorganic lead-free CsSnI_3_-based halide perovskite offers the advantages of enhancing the stability and degradation resistance of devices, reducing the cost of devices, and minimizing the recombination of generated carriers. The simulated standard device using a 1D simulator like solar cell capacitance simulator (SCAPS) with Spiro-OMeTAD hole transporting layer (HTL) at perovskite thickness of 330 nm is in good agreement with the previous experimental result (12.96%). By changing the perovskite thickness and work operating temperature, the maximum efficiency of 18.15% is calculated for standard devices at a perovskite thickness of 800 nm. Then, the effects of replacement of Spiro-OMeTAD with other HTLs including Cu_2_O, CuI, CuSCN, CuSbS_2_, Cu_2_ZnSnSe_4_, CBTS, CuO, MoS_2_, MoO_x_, MoO_3_, PTAA, P_3_HT, and PEDOT:PSS on photovoltaic characteristics were calculated. The device with Cu_2_ZnSnSe_4_ hole transport in the same condition shows the highest efficiency of 21.63%. The back contact also changed by considering different metals such as Ag, Cu, Fe, C, Au, W, Ni, Pd, Pt, and Se. The outcomes provide valuable insights into the efficiency improvement of CsSnI_3_-based PSCs by Spiro-OMeTAD substitution with other HTLs, and back-contact modification upon the comprehensive analysis of 120 devices with different configurations.

## 1. Introduction

Recently, perovskite semiconductors as absorber layers in perovskite solar cell (PSC) devices have drawn wide interest owing to their exceptional optical and electrical properties [1,2,3,4,5,6]. Within a few years, the power conservation efficiency (PCE) of perovskite devices boosted rapidly up to the PCE of 25.8% in 2020 for single-junction perovskite devices, which is comparable with the silicon-based classical photovoltaic devices (PCE of 26.7%) [7]. Further experimental findings showed the maximum PCE of 26.1% in 2021 [8,9], and multi-junction perovskite/silicon tandem solar cells achieved a certified PCE of 32.5% [10]. The cesium tin iodide (CsSnI_3_) is an eco-friendly inorganic solution for energy harvesting solar cell technology [11,12,13,14,15,16,17]. Because of the desirable band gap (E_g_ ≈ 1.3 eV) to achieve the ideal Shockley–Queisser theoretical PCE value (highest efficiency of 33.7% at 1.34 eV) than the band gap of lead-based perovskite absorbers (1.5 eV to 2.3 eV), strong visible light absorption coefficient (≈10^4^ cm^−1^), effective local hole mobility of ∼400 cm^2^/(V·s) for mitigating the recombination rate, small binding energies of Wannier excitons (~10–20 meV), and easy charge separation, the CsSnI_3_ is a more conducive lead-free inorganic perovskite material to achieve the excellent device performance than lead-based hybrid perovskite materials [18,19,20,21].

In 2012, the first reported CsSnI_3_-based PSC measured the efficiency of 0.9% [22]. In 2014, an improved PCE of 2.02% was reported experimentally [23]. The facile oxidation of Sn^2+^ to Sn^4+^, high self-doping effect, and Sn vacancy result in fast degradation of the CsSnI_3_ absorber and instability of CsSnI_3_-based PSCs. Inspiringly, several effective approaches have been suggested to enhance the stability, and quality of CsSnI_3_ thin films, through additive engineering, composition engineering, fabricating the perovskite quantum dots, developing a two-step sequential deposition method [24,25,26,27,28]. An inverted CsSnI_3_-based PSC with efficiency >10%, was experimentally reported in previous literature [29]. These reported PCEs are much lower than the best reported results for lead-based PSCs and the theoretical ideal Shockley–Queisser limit.

Recently, an increasing number of theoretical studies have been carried out to design more efficient inorganic lead-free PSC devices with different electron transporting layer (ETL) and hole transporting layer (HTL) in both normal and inverted structures [30,31,32,33,34,35]. The P-type semiconductors usually regarded as HTLs for extracting the photogenerated holes from the perovskite absorber layer and blocking the carrier recombination. Conventionally, inorganic/organic HTLs have been categorized depending on their chemical structure and contents. Due to great benefits such as the easy and low cost preparation process, high chemical stability, and good hole mobility, inorganic HTLs attracted attention as potential candidates for usage in stable PSCs. Although, organic HTMs usually exhibited higher PCE than inorganic HTMs, their low conductivity and/or hole mobility induce instability of organic HTM-based PSC devices. Therefore, adding dopants is necessary for most of the effective organic HTMs to improve their conductivity and/or hole mobility. Herein, several typical inorganic HTLs such as CuI, CuSCN, Cu_2_O, CuO, MoS_2_ and Cu_2_ZnSnSe_4_ (CZTSe) studied in comparison to organic HTLs such as Spiro-MeOTAD, PEDOT:PSS, P_3_HT, and PTAA to provide some enlightenment for design a more efficient nontoxic lead-free CsSnI_3_-based PSC that can use for further promote PSCs performance, scalable manufacturing, and commercial application using low cost and easy preparation HTMs. Among them, Cu_2_ZnSnSe_4_ (CZTSe) showed the highest efficiency. Cu_2_ZnSnSe_4_ is a p-type semiconductor with high hole mobility (15.1 cm^2^v^−1^s^−1^), low resistivity (0.33 Ω-cm), earth-abundant elemental constituents, and non-toxic properties, which was regarded as a promising inorganic HTL in solar cells [36,37]. The simulated standard device in the present study confirmed the photovoltaic characteristic parameters of the experimentally reported results of the CsSnI_3_-based device with the highest PCE of 12.96%. By optimizing the perovskite absorber thickness, the resultant data proposed a greater thickness for improving the PCE of CsSnI_3_-based device in experiments. According to best of our search, the previous theoretical and experimental studies did not mention this issue. The HTL modification study in this paper is accomplished by considering an optimized thickness for a perovskite layer that was extracted from the agreement between experiment and simulation results. Herein, we focused on the optimization of a CsSnI_3_-based perovskite solar cell with Spiro-OMeTAD HTL as the standard cell. The final generation-recombination rate, current density–voltage and quantum efficiency of optimized simulated standard device with and without Spiro-OMeTAD HTL compared with each other. Then, the effect of Spiro-OMeTAD replacement with other HTLs is investigated.

## 2. Theoretical Methods and Device Structures

Theoretical simulation is a valuable tool for designing and characterizing devices under different conditions. Moreover, simulation reduces the cost and time associated with their experimental investigation by optimizing the necessary parameters and material properties before doing experiments.

### 2.1. Computational Simulation Details

In the present study, the solar cell capacitance simulator structures (SCAPS-1D) software version 3.3.08 is utilized to find the photovoltaic characteristics of short-circuit current density (J_sc_), open-circuit voltage (V_oc_), fill factor (FF), and power conversion efficiency (PCE) by resolving coupled one-dimensional Poisson’s, and continuity semiconductor equations. SCAPS-1D is most popular software among photovoltaic researchers as its simulation results demonstrated a close match with the experimental results [38]. A solar cell device in SCAPS is defined as number of semiconductor layers, in which the transportation of electrons and holes consider by solving the carrier continuity equations and Poisson’s equations iteratively for both electrons and holes. The Poisson’s equation (Equation (1)) relates the charges to the electrostatic potential by following equation:(1)$$-\frac{d^{2}\Psi }{dx^{2}}=\frac{q}{\varepsilon_{0}\varepsilon_{r}}[p\left(x\right)-n\left(x\right)+N_{D}^{+}\left(x\right)-N_{A}^{-}+p_{t}\left(x\right)-n_{t}\left(x\right)]$$

Here, Ψ represents the electrostatic potential, q is the electron charge, ε_0_ represents the permittivity of free space, ε_r_ is the relative permittivity of the semiconductor material, p(n) is the hole (electron) concentration, N_A_^−^ (N_D_^+^) is the density of the ionized acceptors (donors), n_t_ (p_t_) is the trapped electron (hole), and x is the position coordinate. Because of the simultaneous investigation of recombination, generation, drift, and diffusion, the continuity equations are used. The continuity equations for the change in the concentration of electrons (Equation (2)) and holes (Equation (3)) are given as [39]:(2)$$\frac{dn_{p}}{dt}=G_{n}-\frac{n_{p}-n_{p0}}{\tau_{n}}+n_{p}\mu_{n}\frac{dE}{dx}+\mu_{n}E\frac{dn_{p}}{dx}+D_{n}\frac{d^{2}n_{p}}{dx^{2}}$$
(3)$$\frac{dp_{n}}{dt}=G_{p}-\frac{p_{n}-p_{n0}}{\tau_{p}}+p_{n}\mu_{p}\frac{dE}{dx}+\mu_{p}E\frac{dp_{n}}{dx}+D_{p}\frac{d^{2}p_{n}}{dx^{2}}$$
where *G_n_* (*G_p_*) is the electron (hole) generation rate, *n_p_* (*p_n_*) is the electron (hole) concentration in the p-region (n-region), *n_p_*_0_ (*p_n_*_0_) is the equilibrium electron (hole) concentration in the p-region (n-region), *τ_n_* and *τ_p_* denote electron and hole lifetime, *μ_p_* (*μ_n_*) is the hole (electron) mobility, *E* is the electric field, and *D_n_* (*D_p_*) is the electron (hole) diffuse on coefficient.

The photovoltaic parameters such as *PCE*, *V_oc_*, *J_sc_*, and *FF* are related to each other by the following equations [38]:(4)$$FF=\frac{P_{max}}{P_{in}}=\frac{I_{max}\times V_{max}}{V_{oc}\times I_{sc}}$$
(5)$$PCE=\frac{FF\times V_{oc}\times I_{sc}}{P_{in}}$$
where *P*_max_ is the maximum power achievable from the point on the current–voltage (I–V) curve of a solar module under illumination that the product of current and voltage is maximum.

### 2.2. Device Structure

In most of the perovskite device configurations, the perovskite absorber layer is sandwiched between an electron transport layer (ETL), and a hole transport layer (HTL). According to the semiconductor physics, ETL and HTL are responsible for preventing photogenerated charge carriers from recombination and boosting charge transportation by creating the carrier separation paths to deliver electrons and holes from perovskite absorber to the electrodes and go into the external circuit through the metal (Au) contacts.

The simulated CsSnI_3_-based perovskite solar cell in the present study has a normal (n-i-p) structure with TiO_2_ ETL. The exact values of the necessary input parameters for simulation using SCAPS-1D, are difficult to obtain by simulation. The input parameters extracted from literatures are listed in Table 1 [34,40,41,42]. The schematic illustration of device configuration and corresponding energy levels of different layers in simulated device and the band alignment of different HTLs with a perovskite absorber layer are shown in Figure 1.

## 3. Results and Discussions

### 3.1. Photovoltaic Characterization of Standard Device

The photocurrent density–voltage (J–V) and quantum efficiency measurements of the simulated standard PSC device with and without Spiro-OMeTAD recorded using SCAPS-1D software are presented in Figure 2. The fill-factor (*FF*), short circuit current (J_sc_), open circuit voltage (*V_oc_*), and power conversion efficiency (*PCE*) were deduced from the J-V characteristics. A standard simulated AM 1.5 G irradiation system (1000 W.m^−^^2^) and a bias potential scan from 0.0 V to +1.0 V was applied with a scan speed of 100 mV. s^−^^1^ for characterization of PSCs. According to the previous literature, the main responsibility of HTL is effecting on *V_oc_* [41]. Therefore, a disparity appears between series and recombination resistances (R_s_ and R_rec_, respectively) in simulated standard devices with and without HTL. Effect of perovskite thickness modification on photovoltaic parameters of standard optimized device presented in Figure 2. The highest efficiency of 18.06% was obtained for a CsSnI_3_ perovskite layer with a thickness of 800 nm.

By increasing the absorber layer thickness, more amount of generated carriers push to move from the traps and the charge separation ability of perovskite absorber increases. Therefore, the generated current density increases because of the higher rate of producing the photogenerated carriers [33,43,44]. Although, the generation per volume unit decreases (Figure 3a). As shown in Figure 3b, higher increase in the absorber layer thickness results in creating the additional traveling paths for carriers to transfer through the absorber. Therefore, the recombination rate and series resistance decrease by creating a bigger barrier between HTL and ETL. Thus, a thicker perovskite layer produces a weaker built-in electric field, which leads to a reduction of *V_oc_* and *FF* by decreasing the recombination and cell series resistance [45].

By increasing absorber layer thickness, the current density of electron and holes increase (Figure 3c). Although, the total current density in Figure 2b shows a saturated amount for devices contain a perovskite absorber thicker than 800 nm. The results plotted in Figure 3d–f demonstrate energy levels considered fixed and the thickness is the only parameter that is modified in this section.

### 3.2. Photovoltaic Characterization of Standard Device vs. Experimental Results and HTL Free Device

The simulated CsSnI_3_-based standard device with/without HTL compared in Figure 4a. The contact resistance between the metal electrode and perovskite layer causes increasing the series resistance. Therefore, the fill factor and current density of the HTL-free device decrease in comparison to the standard device with spiro-OMeTAD HTL. The HTL acts as a separator layer between perovskite absorber and metal back electrode to enhance the stability and efficiency of perovskite device (Figure 4a). The existence of HTL leads to increasing the amount of collected holes from photo-generated carriers by incident photons at a given energy (QE), and photon absorption at longer wavelengths (Figure 4b). The theoretical characterization results of standard PSC device simulated by SCAPS software at perovskite thickness of 330 nm are in good agreement with the previous experimental report of 12.96% (Figure 4c) [46].

The simulation results in Figure 4c and data listed in Table 2 demonstrate the considered thickness in experimentally fabricated devices is important issue for more efficient CsSnI_3_-based PSC devices. According to the current calculation results, increasing the perovskite thickness from 330 nm to 800 nm improves the efficiency of fabricated devices from 12.96% to 18.11% at room temperature. The thickness of fabricated HTL in devices that can assess HTLs between 300 nm to 800 nm is important for improving efficiency.

The operating temperature for all of devices was considered fixed at 300 K in the present study. The effect of work operating temperature modification on the optimized standard device parameters is investigated. According to the data shown in Figure 5, the highest efficiency of 18.15% is calculated at 320 K for the standard device. By increasing the temperature, the efficiency will decrease rapidly. Because of the increase in the number of charge carriers reaching to the electrodes, a continuous increase in the saturation current is observable. Increasing the reverse saturation current, due to higher surface recombination, causes the *V_oc_* to decline with an increase in temperature, while FF increases up initially, then abruptly starts to decline. The main reason for the fill factor falling is due to the increased resistance of the device.

### 3.3. Photovoltaic Characterization of Simulated Device with HTL Modification

In the simulated CsSnI_3_-based perovskite devices with normal structure, the existence of a hole transport layer helps the photogenerated holes to transfer easily from the absorber layer to metal back contact and also prevent holes recombination with electrons by rapid reflecting them to the circuit through metal back contact. It is noticeable that the input parameters of Spiro-OMeTAD in SCAPS software extract from experiments data. In experiment, the Spiro-OMeTAD is doped with Li-TFSI and tBP dopants for increasing the favorite properties of HTL. Therefore, Spiro-OMeTAD is very expensive. Herein, some proposed p-type semiconductors as HTL in the literature include CuI, CuSbS_2_, CBTS, MoS_2_, MnO_2_, P_3_HT, Cu_2_O, CuSCN, CZTSe, CuO, MoO_x_, PTTA, and PEDOT:PSS investigated to find an appropriate replacement for Spiro-OMeTAD. The resultant J–V curves and external quantum efficiency (QE) for simulated devices with different HTLs and configurations of FTO/TiO_2_/CsSnI_3_/HTL/Au plotted in Figure 6.

Among all of the devices with different HTLs, simulated devices with Cu_2_O, CuSCN, Cu_2_ZnSnSe_4_ (CZTSe), CBTS, CuO, MoS_2_, and MoO_x_ showed better efficiency than Spiro-OMeTAD and the device with Cu_2_ZnSnSe_4_ (CZTSe) as an HTL layer showed the highest efficiency of 21.63% (Table 2) while the minimum PCE achieved for a device with PEDOT:PSS HTL (Figure 6a). The favorable band alignment of Cu_2_O offers a higher current density while lower current density obtained from a device with MoS_2_ HTL owing to unsatisfactory band alignment with perovskite absorber and lower hole transportation to metal electrodes. Therefore, the HTL’s band alignment influences the flow of photogenerated holes in CsSnI_3_-based PSCs. The corresponding QE as a function of wavelength (λ) is studied in the range of 400 nm–1000 nm as shown in Figure 6b. In this study, the ETL, light absorption, and incident photons to the perovskite layer were fixed. The QE started to increase from 400 nm and dropped after 820 nm, corresponding to the band edge of each active material. The maximum QE was obtained for the device with a CZTSe HTL as expected from J–V characteristics and the minimum for the device with a MoS_2_ HTL.

### 3.4. Photovoltaic Characterization of Simulated Device with HTL by Modification of Metal Back Contact

As discussed in the previous section, in the normal structures the photogenerated holes extracted through the metal back contact to the external circuit. Herein, ten different metals of Ag, Cu, Fe, C, Au, W, Ni, Pd, Pt, and Se are used as back contact in simulated devices to investigate the best combination of each HTL for having more efficient devices values among 120 different configurations. The resultant photovoltaic characteristic data are listed in Figure 7. It is noticeable to experimentalists that the fabricated devices with Se, Pt, and Pd back contacts revealed higher efficiency than commonly used Au electrodes for most of the devices with different HTLs. The most characteristic parameter essentially affected by the metal back contact modification in simulated PSCs is the fill factor, due to modification in surface recombination and series resistance created by connection between metal back contact and HTLs.

## 4. Conclusions

The present work reported a comparison between different CsSnI_3_-based PSC devices with modified HTLs and considering different metal back contacts in a normal structure. The simulated devices were characterized by SCAPS software. The standard device with Spiro-OMeTAD HTL showed good overlap with experiment results in 330 nm thickness. The effect of perovskite thickness and working temperature on photovoltaic parameters of devices investigated. Spiro-OMeTAD replaced with ten different HTLs. The device with Cu_2_ZnSnSe_4_ (CZTSe) as an HTL layer showed the highest efficiency of 21.63%. The metal back contact also changed by modifying the work function of the back contact among 120 configurations. Thus, several promising and competitive configurations for high efficient CsSnI_3_-based PSCs are proposed.

## Figures and Tables

**Figure 1 micromachines-14-01562-f001:**
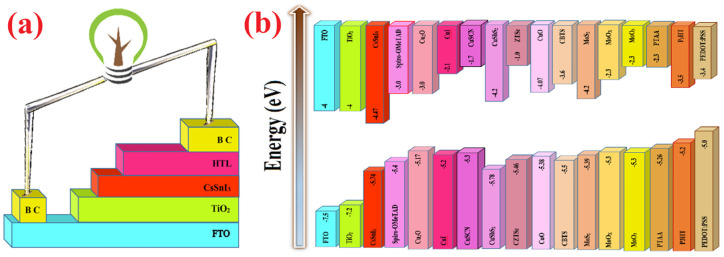
Schematic presentation of (**a**) device configuration, and (**b**) corresponding energy band levels of different layers of simulated inverted CsSnI_3_-based PSCs.

**Figure 2 micromachines-14-01562-f002:**
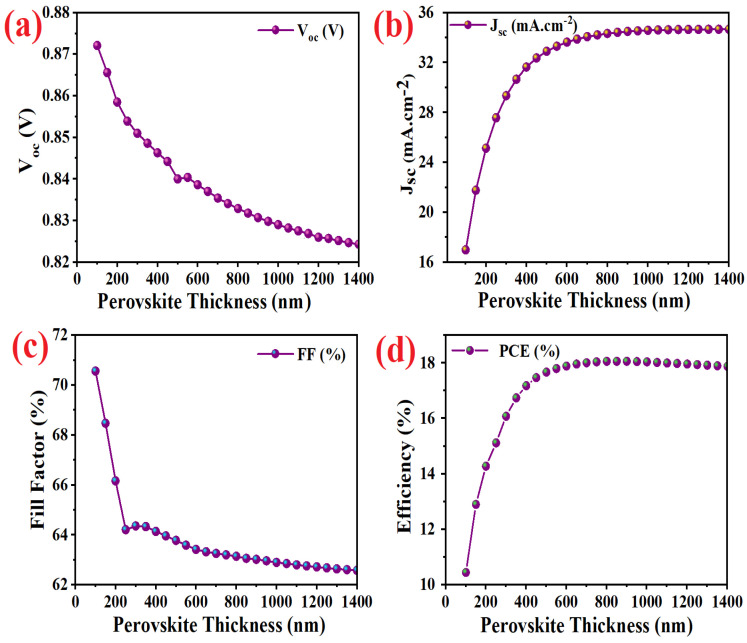
The effect of perovskite thickness modification on photovoltaic parameters of (**a**) current density, (**b**) open circuit voltage, (**c**) fill factor, and (**d**) efficiency of simulated standard devices with Spiro-OMeTAD HTL.

**Figure 3 micromachines-14-01562-f003:**
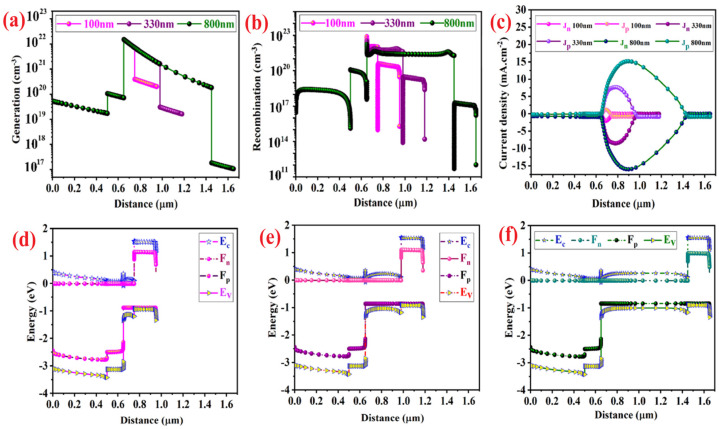
The effect of perovskite thickness modification on (**a**) generation rate, (**b**) recombination rate of photo-generated carriers per volume unit in different layers of simulated devices with different perovskite absorber thickness, (**c**) current density of electrons (J_n_) and holes (J_p_), and conduction band position (E_c_), valence band position (E_v_), Fermi energy levels for device with perovskite thickness of (**d**) 100 nm, (**e**) 330 nm, and (**f**) 800 nm.

**Figure 4 micromachines-14-01562-f004:**
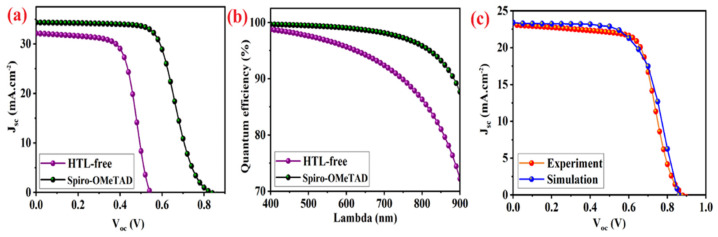
**(a**) The current density–voltage (J-V) curves, and (**b**) quantum efficiency curves of devices with/without Spiro-OMeTAD as the HTL. (**c**) Comparison between J–V curves of simulated device and experimental results.

**Figure 5 micromachines-14-01562-f005:**
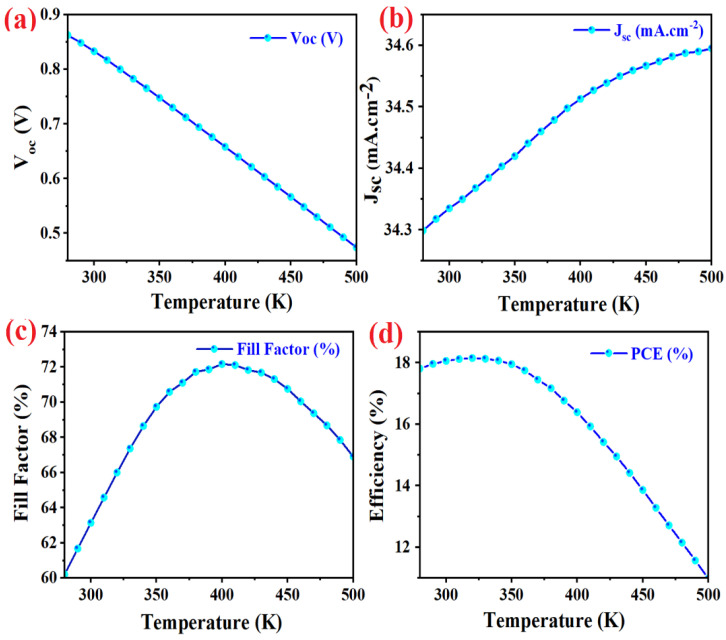
The effect of work temperature on the photovoltaic parameters (**a**) open circuit voltage, (**b**) short circuit current density, (**c**) fill factor, and (**d**) efficiency of simulated standard PSC.

**Figure 6 micromachines-14-01562-f006:**
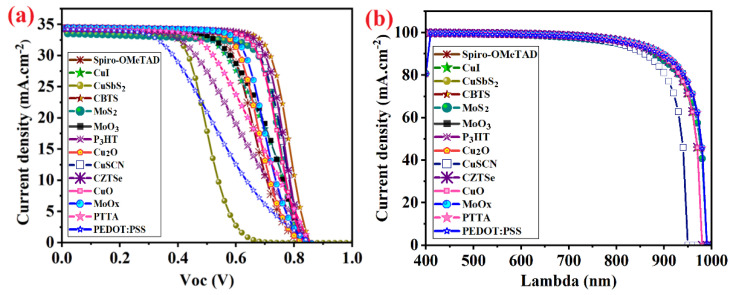
(**a**) The current density–voltage (J–V), and (**b**) quantum efficiency curves of simulated devices with different HTLs.

**Figure 7 micromachines-14-01562-f007:**
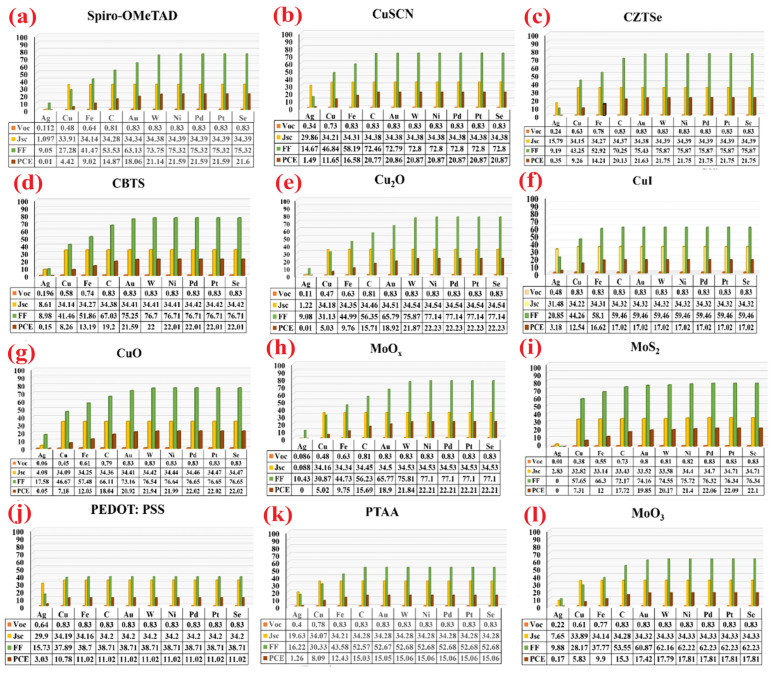
Optimization of photovoltaic parameters of simulated PSCs with different metal back contacts for different HTLs.

**Table 1 micromachines-14-01562-t001:** Input parameters of different layers of FTO, TiO_2_, CsSnI_3_, HTLs for simulated devices in the present study by SCAPS-1D software.

Parameters	N_t_ (1/cm^3^)	N_D_ (cm^−3^)	N_A_ (cm^−3^)	ε	µ_p_ (cm^2^v^−1^s^−1^)	µ_n_ (cm^2^v^−1^s^−1^)	N_v_ (cm^−3^)	N_C_ (cm^−3^)	χ (eV)	E_g_ (eV)	Thickness (µm)
FTO	1.0 × 10^+15^	1.0 × 10^+15^	0	9	10	20	1.8 × 10^+19^	2.0 × 10^+18^	4.0	3.5	0.500
TiO_2_	1.0 × 10^+17^	5.0 × 10^+19^	0	9	10	20	2.0 × 10^+20^	1.0 × 10^+21^	4.0	3.2	0.150
CsSnI_3_	1.0 × 10^+13^	0.0	1.8 × 10^+16^	18	4.37	4.37	1.0 × 10^+19^	1.0 × 10^+19^	4.47	1.27	0.803
Spiro-OMeTAD	1.0 × 10^+15^	0.0	2.0 × 10^+18^	3	2.0 × 10^−4^	2.0 × 10^−4^	1.9 × 10^+19^	2.2 × 10^+18^	3	2.45	0.200
CuSCN	1.0 × 10^+15^	0.0	1.0 × 10^+18^	10.0	25.0	100	1.8 × 10^+18^	2.8 × 10^+19^	1.4	3.8	0.200
Cu_2_O	1.0 × 10^+15^	0.0	1.0 × 10^+18^	7.5	80	200	1.0 × 10^+19^	2.0 × 10^+19^	3.4	2.2	0.200
CuI	1.0 × 10^+15^	0.0	1.0 × 10^+18^	6.5	43.9	100	1.0 × 10^+19^	2.8 × 10^+19^	2.1	3.1	0.200
CuSbS_2_	1.0 × 10^+15^	0.0	1.0 × 10^+18^	14.6	49	49	1.0 × 10^+19^	2.0 × 10^+18^	4.2	1.58	0.200
CZTSe	1.0 × 10^+15^	0.0	5.0 × 10^+16^	9.1	40	145	1.8 × 10^+19^	2.2 × 10^+18^	1.00	4.46	0.200
CBTS	1.0 × 10^+15^	0.0	1.0 × 10^+18^	5.4	10	30	1.8 × 10^+19^	2.2 × 10^+18^	3.6	1.9	0.200
CuO	1.0 × 10^+15^	0.0	1.0 × 10^+18^	18.1	0.1	100	5.5 × 10^+20^	2.2 × 10^+19^	4.07	1.51	0.200
MoS_2_	1.0 × 10^+15^	0.0	1.0 × 10^+17^	13.6	150	100	1.8 × 10^+19^	2.2 × 10^+18^	4.2	1.29	0.200
MoO_X_	1.0 × 10^+15^	0.0	5.0 × 10^+17^	10	2.5	30	2.5 × 10^+17^	3.2 × 10^+16^	2.3	3.2	0.200
MoO_3_	1.5 × 10^+17^	0.0	1.0 × 10^+18^	5.7	1.0 × 10^−4^	1.0 × 10^−4^	4.343 × 10^+19^	4.386 × 10^+19^	2.3	3	0.200
PTAA	1.0 × 10^+15^	0.0	1.0 × 10^+18^	9	40	1	1.0 × 10^+21^	1.0 × 10^+21^	2.3	2.96	0.200
P_3_HT	1.0 × 10^+15^	0.0	1.0 × 10^+18^	3	1.86 × 10^−2^	1.8 × 10^−3^	2.0 × 10^+21^	2.0 × 10^+21^	3.5	1.7	0.200
PEDOT:PSS	1.0 × 10^+15^	0.0	1.0 × 10^+18^	3	4.5 × 10^−2^	4.5 × 10^−2^	1.8 × 10^+19^	2.2 × 10^+18^	3.4	1.6	0.200

**Table 2 micromachines-14-01562-t002:** The photovoltaic parameters of a simulated device with different HTLs.

HTL	*V_oc_* (V)	*J_sc_* (mA.cm^−2^)	*FF*	*PCE* (%)
HTL-free	0.54	32.15	66.58	11.62
Spiro-OMeTAD (exp)	0.86	23.20	65.00	12.96
Spiro-OMeTAD (330 nm)	0.86	23.09	65.02	12.96
Spiro-OMeTAD (800 nm)	0.83	34.34	63.13	18.06
Cu_2_O	0.83	34.51	65.79	18.92
CuI	0.83	34.31	59.46	17.02
CuSCN	0.83	34.38	72.79	20.87
CuSbS_2_	0.72	34.47	53.00	13.23
Cu_2_ZnSnSe_4_	0.83	34.39	75.43	21.63
CBTS	0.83	34.41	72.25	21.59
CuO	0.83	34.41	73.16	20.92
MoS_2_	0.80	33.52	74.16	19.85
MoO_X_	0.83	34.50	65.76	18.91
MoO_3_	0.83	34.32	60.87	17.42
PTTA	0.83	34.28	52.67	15.06
P_3_HT	0.83	34.24	46.44	13.26
PEDOT:PSS	0.83	34.20	38.71	11.02

## Data Availability

All data are included in the manuscript.

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
