# Peer review of "A Comprehensive Study of CsSnI_3_-Based Perovskite Solar Cells with Different Hole Transporting Layers and Back Contacts"

_micromachines, 2023, doi:10.3390/mi14081562_

Round 1
Reviewer 1 Report
The manuscript presents a thorough investigation into the efficiency and stability of CsSnI3-based perovskite solar cells (PSCs), focusing on the impact of different Hole Transport Layers (HTLs) and back-contacts. The authors use a solar cell capacitance simulator (SCAPS) for the simulations, and their approach includes adjustments to the perovskite thickness and operating temperature. A series of HTLs are considered, including Spiro-OMeTAD and a selection of other materials. The authors clearly demonstrate that modifying the HTL and back contact materials can lead to an increase in the device's power conversion efficiency (PCE), with the most efficient HTL being Cu2ZnSnSe4 (CZTSe), resulting in a PCE of 21.63%. This is a significant result that contributes to the ongoing research on lead-free PSCs.
Major Comments
a. More detailed information on the selection process and reasoning behind the chosen HTLs would be helpful. For instance, why were these specific HTLs selected, and how do their properties potentially impact the performance of the PSCs?
b. One area that could be improved is the lack of experimental validation. While the simulation results are promising, experimental confirmation would greatly strengthen the validity of the findings. I recommend the authors to compare their results with experiment, if and where possible.
Minor Comments
a. The manuscript is generally well-written, but there are areas where the language could be made clearer.
b. The abstract could be improved by adding a brief statement on the key results of the study, beyond mentioning the highest efficiency obtained.
Overall, the work provides valuable insights into the efficiency of CsSnI3-based PSCs and demonstrates the potential of alternative HTLs in improving the performance of these devices. The authors are to be commended for their focus on lead-free perovskite solar cells, which is an important area of research in the quest for more sustainable and eco-friendly energy solutions. I recommend that the manuscript be considered for publication after the authors address the comments above, particularly the major ones.
There are areas where the language could be made clearer.
Reviewer 2 Report
Comments and Suggestions can be found in the attached file.

The manuscript's English writing should be improved.
Round 2
